# DNA Methylation-Governed Gene Expression in Autoimmune Arthritis

**DOI:** 10.3390/ijms20225646

**Published:** 2019-11-12

**Authors:** Barbara Brandt, Shima Rashidiani, Ágnes Bán, Tibor A. Rauch

**Affiliations:** 1Department of Medical Biology, Medical School, University of Pécs, Pécs 7624, Hungary; bb.barbara91@gmail.com (B.B.); sh.rashidiani@gmail.com (S.R.); 2Department of Dentistry, Oral and Maxillofacial Surgery, Medical School, University of Pécs, Pécs 7621, Hungary; agnesban2019@gmail.com; 3Department of Biochemistry and Medical Chemistry, Medical School, University of Pécs, Pécs 7624, Hungary

**Keywords:** epigenetics, DNA methylation, autoimmune diseases, rheumatoid arthritis, oral microbiota

## Abstract

Rheumatoid arthritis (RA) is a chronic inflammatory autoimmune disease hallmarked by progressive and irreversible joint destruction. RA pathogenesis is a T cell-regulated and B cell-mediated process in which activated lymphocyte-produced chemokines and cytokines promote leukocyte infiltration that ultimately leads to destruction of the joints. There is an obvious need to discover new drugs for RA treatment that have different biological targets or modes of action than the currently employed therapeutics. Environmental factors such as cigarette smoke, certain diet components, and oral pathogens can significantly affect gene regulation via epigenetic factors. Epigenetics opened a new field for pharmacology, and DNA methylation and histone modification-implicated factors are feasible targets for RA therapy. Exploring RA pathogenesis involved epigenetic factors and mechanisms is crucial for developing more efficient RA therapies. Here we review epigenetic alterations associated with RA pathogenesis including DNA methylation and interacting factors. Additionally, we will summarize the literature revealing the involved molecular structures and interactions. Finally, potential epigenetic factor-based therapies will be discussed that may help in better management of RA in the future.

## 1. Introduction

It is well established that gene expression is defined by cis-regulatory elements and trans-acting factors, and both of them can be directly or indirectly affected by epigenetic factors [1]. Cis-acting elements are short consensus DNA sequences in the promoter regions and distant regulatory elements (e.g., enhancers and silencers) are recognized by trans-acting factors (i.e., transcription factors). Epigenetic factors are involved in establishing such a chromatin milieu around cis-elements that establish favourable or refractive environment for gene transcription [2]. Importantly, epigenetic machineries have connections with environment through signalling pathways, and gene expression profile is shaped by epigenetic factors to provide the best cellular response to the actual stimulus [3]. Epigenetic signals are covalent but reversible modifications on genomic DNA and histone molecules. DNA methylation is the addition of a methyl group to the fifth position of cytosine occurring at CpG dinucleotides, which is catalysed by DNA-methyltransferases (DNMTs) that transfer a methyl group from S-adenyl methionine (SAM) to the fifth carbon of a cytosine residue to form 5-methylcytosine (5mC). In mammals, approximately 98% of DNA methylation might occur in clusters of GC-rich sequences called CpG islands (CGIs) [4]. DNA methylation is involved in regulatory processes including gene expression, chromatin structure, genomic imprinting, transposon silencing, and X-chromosome inactivation. CGIs often overlap with promoters and first exons of genes and also in regions toward the 3′ end [5]. CGIs are frequently evolutionary conserved sequences that promote gene expression by regulating the chromatin structure and transcription factor binding. The methylation of CGIs results in gene silencing since it can interfere with transcription factor binding and might recruit repressive methyl-binding proteins [5]. DNA methylation also occurs in inter- and intragenic regions, which plays a crucial role in the inactivation of transposable elements and silencing of cryptic promoters [6].

We will discuss RA-associated methylomes (i.e., in synovial fibroblasts and immune cells); DNA methylation machinery and its interacting epigenetic factors; and their involvement in disease initiation, promotion, and progression. The emerging role of microbiome and its interaction with epigenetic machinery in autoimmune arthritis is going to be addressed, which might be a key process in RA aetiology.

## 2. Writing, Interpreting, and Erasing DNA Methylation

For practical use; epigenetic enzymes that *establish*, *recognize*, and *remove* DNA methylation can be divided into three groups based on their catalytic activity; accordingly, there are writers, readers, and erasers. Writers can catalyse the formation of 5mC, readers are able to recognize and bind to 5mC resulting in the regulation of gene expression, and finally erasers modify and remove the methyl group of 5mC (Figure 1) [7].

### 2.1. “Writing” the DNA Methylation

Epigenetic reprogramming events take place during the mammalian development, and they play an important role in writing the DNA methylation after the implantation of the blastocyst [8]. A wave of de novo methylation also occurs during cellular differentiation and it is mediated by the DNMT3A and DNMT3B enzymes that are capable of methylating DNA without any preference (Figure 2). Another protein called DNMT3-like (DNMT3L) is also involved in the de novo methylation processes, but it has no catalytic activity [7,9]. Even if DNMT3L does not have catalytic activity, it plays an important role in regulating de novo methylation by interacting with DNMT3A and DNMT3B thus increasing their methyltransferase activity. DNMT3A is expressed ubiquitously while DNMT3b has low expression among the differentiated tissues. The knockout *Dnmt3b* mice are embryonic lethal; thus, this enzyme is probably required during early development. *Dnmt3a* knockout mice are runted and survive to approximately 4 weeks after birth suggesting that Dnmt3a is required for normal cellular differentiation [7,10]. The DNMT3A and DNMT3B catalytic activity and specificity are controlled by specific histone modifications. The DNMT3A and DNMT3B proteins are very similar in structure and function and are consist of a C-terminal catalytic domain and an N-terminal regulatory domain, which further contains a PWWP (proline-tryptophan-tryptophan-proline) domain responsible for DNA-binding, and an ADD (ATRX-DNMT3-DNMT3L) domain [7,8,11,12]. The ADD domain has two C_4_-type zinc fingers, which can interact with the N-terminal tail of H3 with unmodified lysine 4 (H3K4me0) [13]. The ADD domain also interacts with its own catalytic domain; thus, it can block the DNA-binding affinity. Unmodified histone H3 can disrupt the catalytic domain-ADD domain interaction resulting in the reactivation of the DNMT3A [14].

### 2.2. Maintaining the DNA Methylation

The heritability of DNA methylation patterns is due to the maintenance of DNA methylation, which contributes to the cellular memory [8]. During DNA replication, the DNA becomes hemimethylated since the newly synthesized daughter strand is unmethylated, while the parental strand remains methylated. To maintain the methylation, a DNMT enzyme recognizes the hemimethylated DNA strand and methylates the DNA on the daughter strand. The major maintenance methyltransferase is a 200-kDa protein named DNMT1, which is specific to CpG nucleotides and constitutively expressed in proliferating cells [8,15]. The DNMT1 is upregulated during the S-phase of the cell cycle and its activity is coupled to DNA replication [16]. Disruption of DNMT1 in mouse embryonic stem cells caused a global loss of CpG methylation [8]. The DNMT1 is composed of an N-terminal regulatory domain and a C-terminal catalytic domain, which contains highly conserved DNA methyltransferase motifs (Figure 2). The N-terminal region of the DNMT1 has unique domains such as the DNA binding CXXC domain, the bromo-adjacent homology (BAH) domain, the proliferating cell nuclear antigen (PCNA) binding domain (PBD), and also the replication foci-targeting sequence (RFTS). The RFTS contains a ubiquitin-interacting motif (UIM), which can recognize the ubiquitinated histone H3 at lysine 18 (H3K18ub) that provides a docking site for DNMT1 targeting the replication foci [9,14,17]. Beside the DNMT1, a 95-kDa protein called UHRF1 plays an important role in maintaining the DNA methylation in mammals (Figure 3C). The UHFR1 is an ubiquitin E3 ligase, and it is also upregulated in S-phase and localized at the DNA replication foci. The deletion of UHFR1 in mice caused embryonic lethality and genome-wide DNA hypomethylation [18]. UHRF1 contains five functionally distinct domains including the ubiquitin-like domain (UBL), the tandem tudor domain (TTD), the plant homeodomain (PHD), the SET and RING associated domain (SRA), and the Really Interesting New Gene domain (RING). The SRA domain binds hemimethylated CpG dinucleotides and tends to play an important role in linking DNMT1 to the newly synthesized DNA [19,20]. The TTD and PHD domains act in combination to read the histone H3 trimethylated at lysine 9 (H3K9me3), which is a transcriptionally repressive chromatin mark mostly present in the pericentric heterochromatin [21]. The RING domain possesses E3 ubiquitin ligase activity, which can ubiquitinate histone H3 at lysine 18 (H3K18ub) that serves as a binding site for DNMT1 [17].

### 2.3. “Reading” the DNA Methylation

The DNA methylation is recognized by the “reader” molecules that have a high affinity for 5mC and generally inhibit transcription factor binding, thus reducing the rate of transcription [8,15]. Three separate families of MBPs have been discovered: the methyl-CpG-binding domain (MBD) proteins, the ubiquitin-like with PHD and RING Finger (UHRF) domain proteins, and the methyl-CpG binding zinc finger proteins [7]. The first family of DNA methylation readers has a conserved methyl-CpG-binding (MBD) domain that has a great affinity for single methylated CpG sites [7]. Based on their specific features and functions, they are separated into three subclasses: MeCP2-MBD, HMT-MBD, and HAT-MBD (Figure 3). The MeCP2-MBD subfamily contains seven different proteins including MeCP2, MBD1, MBD2, MBD3, MBD4, MBD5, and MBD6. The MeCP2 is a 50 kDa multidomain protein, which can bind selectively to a single methylated CpG [22]. The MeCP2 has an N-terminal MBD and C-terminal transcriptional repression domain (TRD) [23] MeCP2 cooperates with the Sin3A co-repressor and histone deacetylases (HDAC1 and HDAC2) to alter chromatin structure and therefore repress transcription [8]. In addition, it has been shown that MeCP2 can increase chromatin condensation by recruiting histone methyltransferases that cause H3K9 methylation, thus decreasing the rate of transcription [7]. Mutations of the MeCP2 gene are the main cause of the Rett syndrome and are involved in several neurodegenerative diseases. Mecp2 mutant mice show abnormalities in the number of synapses and in neuronal morphology, which confirms the suggestion that MeCP2 plays a crucial role in regulating the gene expression [24]. Recently it was supposed that Rett syndrome is an autoimmune disease [25], which is a hypothesis based on a number of publications [26,27]. This hypothesis is in line with those findings that describe implication of MeCP2 in other autoimmune diseases including rheumatoid arthritis (RA).

The MBD1 is a 55 kDa multidomain protein which also contains a conserved N-terminal MBD domain and a C-terminal TRD domain. In between these domains, the MBD1 protein further has two or three CXXC-type Zinc fingers (CXXC1, CXXC2, CXXC3), which number depends on the alternative splicing events [28]. Isoforms containing all the three CXXC-type Zinc finger motives are capable of binding to unmethylated DNA as well; therefore, these proteins are able to repress the transcription not only of methylated regions but also of unmethylated sequences. Isoforms containing the first two motifs can only bind to methylated CpGs and decrease the rate of transcription [29]. MBD1 can promote heterochromatin formation. Altered expression of MBD1 was detected in various cancer types [30] and in autoimmune diseases such as SLE [31].

The MBD2 protein is a transcriptional repressor protein; MBD2 is unique among the MeCP2-MBD subfamily members in as much as highly expressed in spleen, which is a pivotal organ for innate and adaptive immunity as well [32]. MBD2 is an integral component of nucleosome remodeling and histone deacetylation (NuRD) complex, which regulates gene transcription and genome and cell cycle progression [33]. As part of the NuRD complex, MBD2 is involved in B and T cell maturation, which might explain that MBD2 KO mice develop lupus-like syndrome [34]. In addition to playing a role in DNA hypermethylation, MBD2 is also involved in DNA demethylation by interacting with TET2 [35], which is one of the main acting erasers of DNA methylation in somatic cells. (Discussed later).

Human MBD3 has a point mutation in its MBD domain, therefore, this protein loses its binding affinity for methylated CpG sequences [22]. On the other hand, this protein is able to bind to 5-hydroxymethylated cytosine (5hmC), and it has a role in the maintenance of 5hmC by colocalizing with the DNA demethylase TET1 [36].

MBD4 can bind to methylated DNA due to its MBD domain and it also contains a C-terminal thymine DNA glycosylase domain that can selectively remove T from T-G mismatches in CpG sites [8,30]. Differential expression of MBD4 was detected in multiple sclerosis, which is an autoimmune disease affecting the myelin sheet of neurons [37].

The least characterized two members of the MeCP2-MBD subfamily are the MBD5 and MBD6 proteins. Although these proteins are involved in epigenetic regulation [30], their MBDs cannot bind 5mCs [38]. Therefore, the molecular function of MBD5 and MBD6 in DNA methylation is elusive.

The additional two subfamilies (i.e., HMT-MBD and HAT-MBD) of methylated DNA binding proteins cannot bind 5mC similarly to MBD5 and MBD6. Instead, they play role in epigenetic modification of histones by their methyltransferase activity (i.e., SETDB1 and SETDB2) or act histone acetyltransferases (i.e., BAZ2A and BAZ2B) [22]. Although MBDs of these proteins are evolutionarily conserved, their molecular functions have not been clarified.

### 2.4. The UHRF Proteins or SRA Domain-Containing Proteins Have Two Members

The UHRF proteins or sra domain-containing proteins have two members: UHRF1 and UHRF2. Both of them have five distinct domains, described above alongside with the function of UHRF1. The UHRF2 is also known as NIRF or Np97, and it has a role in the regulation of cell cycle. UHRF2 can also enhance the enzymatic activity of TET1 and can bind to 5hmC [22].

### 2.5. The Methyl-CpG Binding Zinc Finger Proteins

The Methyl-CpG Binding Zinc Finger Proteins are a large family of MBPs, and all the members contain C-terminal Zinc finger motifs, although they can bind to methylated and unmethylated DNA sequences as well. This family consists of 8 members, including Kaiso (also known as ZBTB33), zinc-finger, and BTB domain containing 4 (ZBTB4), ZBTB38, ZFP57, Krüppel-like factor 4 (KLF4), early growth response protein 1 (EGR1), Wilms’ tumor 1 (WT1), and CCCTC-binding factor (CTCF) [39].

Kaiso is a transcription factor that can bind both methylated and unmethylated DNA sequences and is associated with transcriptional repression [22]. This protein plays an important role in directing the nuclear receptor corepressor complex (N-CoR) to methylated and unmethylated DNA regions, therefore, promoting heterochromatin formation. [40] Kaiso can regulate the stability of cadherins as well; thus, it plays a role in cell adhesion [41].

ZBTB4 can bind to methylated and unmethylated DNA as well, and it functions as a transcriptional repressor protein since it can recruit Sin3/histone deacetylases and promote the silencing of the target gene [42].

ZBTB38 can bind to a single methylated CpG and to unmethylated DNA as well and plays a role in cell proliferation and differentiation. ZBTB38 can also regulate gene expression, DNA replication, and genomic stability. Notably, both ZBTB4 and ZBTB38 are highly expressed in the brain [7]. ZBTB38 plays a role in IL1R2 downregulation in autoimmune arthritis [43].

ZFP57 can bind to methylated CpG sites, and it is associated with several cellular processes including regulation of gene expression, genomic imprinting, and cell signalling [22]. Beside the zinc finger domain, ZFP57 also has an N-terminal Krueppel-associated box (KRAB) domain. KRAB interacts with KRAB-associated protein 1 (KAP1), which recruits several heterochromatin-inducing factors including histone methyltransferases; thereby, ZFP57 is associated with transcriptional repression [44].

The KLF4 protein is involved in the genomic stability, cell proliferation, regulation of cell cycle, DNA damage response, and apoptosis. Uniquely among the DNA methylation readers, it functions as a transcriptional activator since it can bind to methylated CpG sequences and recruit chromatin remodelling complex [45].

The stress signal induced EGR1 transcription factor plays a role in differentiation, inflammation, wound healing, and the maintenance of synaptic plasticity. It can bind to cytosines independently of their methylation status- in a GCC(T/G)GGGCG consensus sequence near the promoter of the regulated gene [22].

The WT-1 tumour suppressor protein is involved in the regulation of cell growth, differentiation, cell cycle, and cell division, and it recognizes the same consensus sequence near the target gene as the EGR1 protein, regardless of its methylation status [22].

The CTCF transcription factor can bind to both methylated and unmethylated DNA sequences since it has 11 conserved zinc finger DNA-binding domains [46]. This protein is able to inhibit the interaction between the promoter and enhancer region and plays a role in several cellular processes including regulation of chromatin structure and gene expression [47].

## 3. “Erasing” the DNA Methylation

Removal of methyl group from 5mC residues (i.e., DNA demethylation) can occur both replication-independent (active) and replication-dependent (passive) ways [48]. The passive demethylation takes place when DNMT1 does not catalyse methyl group addition onto the hemimethylated DNA template following replication. On the other hand, active DNA demethylation is independent of replication and occurs in both dividing and non-dividing cells, and it is associated with the modification and removal of 5mC by TET proteins (Figure 4) [49,50]. TET proteins are named after the ten-eleven translocation (t(10;11)(q22;23)), which is present in acute myeloid and lymphocytic leukaemia, resulting in a fusion between the mixed-lineage leukaemia 1 (MLL1) gene on human chromosome 10 and TET1 gene on human chromosome 11 [51]. So far, there is no known mechanism in mammalian cells that cleaves the strong covalent bond that binds cytosine to a methyl group [7]. Instead, TET proteins are able to oxidize 5mC into 5-hydroxymethylcytosine (5hmC), 5-formylcytosine (5fC) and 5-carboxycytosine (5caC) [48]. The oxidized derivatives of 5mC are recognized and eventually removed by certain DNA repair enzymes, which then introduce an unmodified cytosine into the DNA strand [48]. Some studies argue that activation-induced cytidine deaminase (AICD) [52] is implicated in demethylation conducting cytidine deamination followed by mismatch repair. However, another study showed that 5hmC, 5fC, and 5caC could be directly removed by DNA dehydroxymethylases, deformylases, and decarboxylases [53]. Interestingly, there are data supporting that 5hmCs hydroxyl group can be removed by DNMT enzymes, directly converting 5hmC to cytosine in vitro. The exact mechanism of TET-mediated DNA demethylation is still not quite clear [51]. It is possible, that more than one pathway is implicated, and different cells might use different pathways determined by whether demethylation is global or local in the genome [52].

Genome-wide-profiling methods showed that 5hmC is enriched at promoters resulting in a decreased gene expression in human and mice embryonic stem cells and mouse neural progenitor cells. On the other hand, 5hmC is also enriched at gene bodies and is positively correlated with gene expression. Furthermore, in all mammalian cells, 5hmC is enriched at enhancers and is associated with gene activation [51].

The identification of TET enzymes has been considered as one of the most significant discoveries for current epigenetics. TET enzymes provide a mechanistic explanation for active DNA demethylation, which was a hypothetical pathway for a long time. All three of the TET proteins have an N-terminal regulatory domain and a C-terminal catalytic domain, which further contains a cysteine-rich region and a double-stranded β-helix (DSBH) domain (Figure 4) [54,55]. The N-terminal regulatory domains of TET proteins are not conserved, TET1 and TET3 have a CXXC zinc finger domain in their N-terminus, which can selectively bind to unmodified CpGs and has been shown to be present in multiple chromatin-interacting proteins including CFP1, MLL, MBD1, and DNMT1 [52]. TET1′s CXXC domain is able to bind to unmodified, 5mC-modified and even to 5hmC-modified CpG sequences while TET3′s CXXC domain only binds to unmodified cytosines.

## 4. Pathogenesis of Rheumatoid Arthritis (RA)

RA is a complex inflammatory autoimmune disease that primarily affects synovial joints and gradually destroys articular surfaces, additionally, extra-articular manifestations can also occur in advanced RA attacking the skin, lung, heart and eyes [56]. A growing body of evidence shows that pathogenesis of early and late RA stages are quite different and accordingly do not involve the same pathomechanisms (reviewed in [57]). During the early period (i.e., around the onset of the disease and a following 4–6 months) immunological factors and mechanisms are decisive and, therefore, more sensitive to inflammatory mediator-focused biotherapies. Later, in the advanced/refractory period, RA can be regarded as genetic and epigenetic factor driven disease, which generates a new field for novel therapeutic options including druggable epigenetic factors.

Approximately 1% of the human population is affected by RA, but significant alterations in prevalence can be observed in different ethnic groups [58]. The aetiology of RA remains unknown, and therefore, there is a long-standing uncertainty regarding the nature of the involved factors [59]. The question to be answered was whether genetic or epigenetic factors dominate and determine RA pathology. During the last 10–15 years, genome-wide association studies (GWASs) have identified a number of potential RA-linked risk alleles. GWASs have been identified more than 100 susceptible loci, and most of the risk-associated single nucleotide polymorphisms (SNPs) were mapped into non-coding regions of the human genome [60]. To interpret these findings, it must be supposed that RA-risk SNPs could hit gene regulatory regions by altering transcription factor binding sites or potential DNA methylation targets (i.e., CpG dinucleotides) [15,61].

Epigenome-focused studies revealed disease-associated DNA methylation and histone modification profiles in promoter and enhancer regions [62]. Regarding the genetic vs. epigenetic aetiology debate, it was a significant observation that concordance rate is low in monozygotic twins relative to dizygotic twins [63], which argues for the epigenetic origin of autoimmune arthritis. Familial accumulation of RA is also observed, implying that both genetic and environmental components can be implicated in RA aetiology. Although limited information is available about all of the involved epigenomes, recent studies suggest that pools of genes showing altered epigenetic profiles are likely involved in the development of the disease [59,64]. However, neither genetic nor epigenetic risk factors act separately; rather, they are parts of complex networks controlling critical cellular functions.

RA pathology in joints has at least two major involved components, one is represented by the infiltrating/invading immune cells including T- and B-lymphocytes, while the other one is comprised by synovial cells producing lubricant for the joint. T cells produce proinflammatory cytokines including tumour necrosis factor (TNF), various interleukins (ILs) and growth factors. B cells are the source of autoantibodies, for example rheumatoid factor (RF) and anti-citrullinated protein antibody (ACPA) [59,65]. In healthy joints, synovial fibroblasts are several cell layer thick lining membranes (i.e., synovium) surrounding the joint cavity filled by synovial fluid. However, during RA pathogenesis, synovium is heavily infiltrated by immune cells leading to multilayer thickened synovial tissue (i.e., pannus), which aggressively attack the articular structure. RA synovial fibroblasts (RASFs) produce cytokines, chemokines, and various proteases that ultimately destroy the adjacent articular cartilage and bone [66].

RA-associated epigenetic changes have been investigated both in immune cells isolated from peripheral blood samples and RA synovial fibroblasts (RASFs) that were collected after total joint replacement surgeries. RASF-related epigenetic data were usually compared to data gained from synovial fibroblasts from osteoarthritic (OASF) joints [62]. Different animal models of RA were also employed for exploring epigenetic alterations in initiation and progression of autoimmune arthritis [67,68]. Historically, the first autoimmune diseases affecting global DNA methylation changes were observed in the context of RA [69], next, analyses were conducted at single gene level, and finally whole genome (i.e., methylome) mapping studies were carried out. Combination of next generation sequencing and novel bioinformatics revolutionized arthritis research by revealing a number of druggable genes [70].

### 4.1. DNA Methylation Profiling in Synovial Fibroblasts

Steffen Gay’s lab in Zürich conducted one of the first epigenetic studies on RASF, which was focused on a transposable element of the human genome (i.e., Line-1) that is silenced by DNA methylation in the promoter region [71]. They found that increased expression of LINE1 in RASF was due to a global DNA hypomethylation is RASF [71]. Similar global hypomethylation could not be detected in OASFs, suggesting that global DNA hypomethylation is a disease-specific event in RA pathology. Later epigenetic studies were focused mostly on non-repetitive elements of the human genome rather than unique sequences, which were dominantly differentially methylated promoters. These studies revealed DNA hypermethylation in promoter region of death receptor gene 3 (DR3), which contributed to disease-specific downregulation of gene expression [72]. DR3-related study can explain RASFs resistance to apoptosis and aggressive transmigration ability [73]. RA-related DNA hypomethylation was also detected in RASFs; CXCL12 gene’s promoter methylation level was lower than in OASFs, and accordingly, its expression was higher [74]. CXCL12 gene encodes a well-known chemokine, which is an attractant for immune cells, and in addition, it induces overexpression of matrix methaloproteinases (MMPs) that mediate joint destruction [75]. The above presented two hypothesis driven epigenetic studies contributed definitely to the better understanding of RA pathogenesis; however, with the advent of genome-wide analysis tools, the search for disease-specific alterations and gene expression profiles became more effective. In the first period of the genomic era, various microarray platforms were combined with DNA methylation, chromatin immunoprecipitation (ChIP), and gene expression detection methods [76]. Later, next-generation sequencing (NGS) replaced microarrays, which made it feasible to explore the whole genome and all genes (and their alternative transcripts) in a single experiment [70,77]. The best example for the power of this approach was provided by Ai et al., demonstrating that RA pathogenesis might be different from joint to joint [62]. More specifically, RASFs isolated from knee and hip joints were different regarding their DNA methylation pattern and gene expression profiles, which explains the diversity of drug responses in RA patients and implies the significance of personal medicine. However, it is still a fascinating question whether RAFS are imprinted somewhere outside of the joints before migrating into the joint or whether they gain the characteristic DNA methylation pattern after arriving.

The most comprehensive epigenetic analysis in the context of RASF was conducted by Gary Firestein’s lab; their analyses included the survey of six histone modifications, open chromatin, RNA expression, and whole genome methylation profile [70]. By comparing RASF and OASF data, 13 differentially modified epigenetic regions (DMERs) were identified. Nine of the DMERs overlap with enhancer regions; the other four regions were associated with promoters [70]. This study proved that transcriptional regulatory regions (i.e., enhancers and promoters) make the difference between RA and OA. Further analysis of DMERs’ related genes new (and unexpected) biological pathways were revealed for RA pathogenesis including p53 signalling, protein kinase A, and Huntington’s disease signalling.

### 4.2. DNA Methylation Profiling of Immune Cells

The other major contributors to RA pathology are the infiltrated immune cells. Similarly to RASFs, the first observed epigenetic alteration that could be associated with RA pathogenesis was global hypomethylation [69]. It is well-documented that altered DNA methylation profiles, including genome-wide hypomethylation and promoter-specific de novo hypermethylations have been implicated in carcinogenesis [6,78]. Similar DNA methylation pattern changes have been observed in an experimental model of autoimmune arthritis [43,79]. A number of differentially methylated promoters were discovered by hypothesis driven studies including IL6, IL10, and IL1R2 promoters. Investigation of these genes was the obvious choice since they had emerged from traditional immunological studies. Accordingly, IL6 is a pro-inflammatory cytokine that stimulates inflammation and contributes to RA pathogenesis [80]. It turned out that in control (healthy) peripheral blood mononuclear cells (PBMCs) IL6 promoter is hypermethylated compared to cells isolated from RA patients [81]. The significance of IL6 promoter methylation is underpinned by the fact that targeted neutralization of IL6-based therapies is the most effective RA treatment option [82]. IL-10 promoter was differentially methylated in arthritic cells [83], which is an anti-inflammatory cytokine and inhibits pro-inflammatory cytokine (e.g., IFN-γ, IL-2, IL-3, TNFα) synthesis. IL1R2 is a decoy receptor, which can interfere with IL1-mediated inflammation. It was found that IL1R2 promoter was hypomethylated in RA patients [84]. In an animal model of RA, it was dissected how an arthritis-specific DNA hypomethylation event can support autoimmune arthritis by interfering with an anti-inflammatory pathway [43]. It was revealed that ZBT38, a transcriptional repressor with methylated DNA binding ability, was hypomethylated in B cells, which was accompanied by increased ZBTB38 expression. Accordingly, the upregulated ZBTB38 could bind to the promoter region of IL1R2 gene that resulted in low expression of IL1R2 gene and ultimately led to compromised anti-inflammatory action (i.e., neutralization of IL1B) [43].

Information about RA-related genome-wide B cell DNA methylation is limited and mostly relies on studies conducted on PBMCs. These studies described characteristic global DNA hypomethylation [85,86] and identified locus-specific hypomethylation of certain promoters [83,87].

Arthritic B cell methylome-focused study revealed a novel gene regulatory cascade that is affected in autoimmune arthritis in mice. Aryl hydrocarbon receptor (AhR) is a transcription factor that supervises the expression of a number of genes, and depending on the interacting factor, AhR can act either as an activator or repressor [88,89]. One of AhR’s primer target is the activation induced cytidine deaminase (AICDA) gene that encodes an enzyme that plays multiple roles in germinal center formation, wherein antibodies are generated [90,91]. In the absence of AhR, AICDA expression is high and antibody production (including autoantibodies) elevated, which contributes the gradual destruction of synovial joints. AhR is epigenetically silenced by DNA methylation in arthritic B cells; thus AICDA gene expression is high and accordingly GC formation, and antibody production is elevated [79]. These AhR-related findings initiated human studies that identified a differentially methylated region (DMR) upstream from AHR promoter. ChIP-Seq data demonstrate that a number of transcription factors [92,93] can bind to this DMR, which suggests that this region is a B cell-specific enhancer regulating AhR expression.

From epigenetic point view, regulatory T cells (Tregs) are the most characterized immune cell types in autoimmune pathogenesis. Tregs play multiple roles in the maintenance of immune homeostasis that is mainly attributed to FOXP3, a master transcription factor, which is tightly regulated by epigenetic mechanisms. There are intronic enhancers and promoters in the FOXP3 locus that are bound by specific transcription factors. Methylation status of regulatory elements is determined by the cooperation of DNMTs [94], MBD2 [35], and MeCP2 [27] epigenetic factors. For stable expression of Foxp3 in Tregs, it is critical that an evolutionarily conserved intronic enhancer must be demethylated, whose hypomethylated state enables binding of linage specific transcription factors and subsequent activation of FOXP3 promoter [95,96,97]. It was demonstrated that demethylation factors (i.e., TETs) play an important role in the actual methylation status of intronic regulatory elements [98,99]. The well-characterized immunological and epigenetic backgrounds of Tregs made tempting to use them for preventing excessive immune responses and autoimmunity. However the ongoing clinical trials will be essential to finely tune Treg dose, the timing, and the immunosuppressive regimen [100].

## 5. Oral Microbiota Might Predispose Host to Autoimmune Diseases by Promoting Epigenetic Changes

Human’s second-best friend is the microbiota (i.e., community of the symbiotic microorganisms) that lives with the host for thousands of years and can influence all aspects of life by metabolites and overexpressed enzymes. The diet of the host directly influences the microbiome and in turn can alter the epigenetic milieu in the neighbouring cells, which might induce systemic changes later.

Periodontal disease is a known risk factor of RA, and it has been observed that the oral and gut microbiota are different in treatment naïve and chronic RA patients compared to control individuals with no systemic diseases [101,102]. Studies suggest that certain member(s) of microbiota community might be implicated in pathogenesis of RA. *Porphyromonas gingivalis* (*P. gingivalis*) is a pathogenic member of the oral microbiota, which possesses an enzyme called peptidyl-arginine deiminase (PPAD) that can catalyse citrullination of proteins [103]. Human genome encodes five similar enzymes with equivalent activity in a gene cluster on chromosome 1 [104]. PADI4 is an epigenetic factor, which as the only member of human PADI family is translocated into the nucleus and implicated in histone code writing [105,106]. GWASs and meta-analyses have revealed PADI4-associated polymorphisms that confer susceptibility to RA in various human populations [107]. RA evolves over multi-year period with no symptoms (i.e., preclinical RA); however, anti-citrullinated protein antibodies (ACPA) can be detected in RA patients, which is highly predictive for the onset of disease [108]. The hypothesis regarding the pathological connection between periodontitis and RA is that the breakdown of immune tolerance is due to PAD- or PPAD-dependent citrullination in periodontal tissues (Figure 5). It is proven that *P. gingivalis* is associated with disease initiation and could be target for preventive interventions in RA [109,110]. Recent studies identified *Cryptobacterium curtum* (*C. curtum*), an anaerobe Gram-positive bacterium, whose organism is capable of producing high amounts of citrulline that might initiate RA pathogenesis [111]. Eradication of biofilm inhabitant *P. gingivalis* by antiseptics or/and antibiotics is not efficient [112]. New therapies on the horizon include antimicrobial peptides that reduce *P. gingivalis* titer but are not toxic for human monocytess and gingival fibroblasts [113]. It is tempting to hypothesize that local application of PADI4 inhibitors might be also effective against *P. gingivalis.*

## 6. Epigenetic Enzyme-Based Therapies

With the lack of proper cure for RA, the goals of treatments are complex, including mitigation of pain, reducing or halting joint destruction, and preventing extra-articular complications. Therefore, in most cases, the optimal disease treatment can only be achieved through combined application of different drugs [110]. When a combination of traditional disease-modifying anti-rheumatic drugs (DMARDs) and non-steroid anti-inflammatory drugs (NSAIDs) and/or low-dose glucocorticoids does not provide satisfactory response, highly specific biologics are introduced into the treatment regimen. Classical DMARDs and NSAIDs are usually small molecules that broadly affect many components of the immune system, whereas rationally designed biologics (e.g., anti-TNFα, anti-IL-1β, IL-6 antagonist) specifically inactivate/neutralize key mediators of inflammatory processes [114]. There is an obvious need to discover new drugs for RA treatment that have different biological targets or modes of action than the currently employed therapeutics. Epigenetics opened a new field for pharmacology, and DNA and histone methylation-implicated factors are feasible targets for RA therapy. The FDA has approved two DNA methylation inhibitors (AzaC and AzadC) for use in cancer chemotherapy, which induce radical epigenome remodelling by capturing DNA methyltransferases (DNMTs). However, in the case of arthritis, using less radical treatments, including highly specific DNMT-blockers, might be sufficient to reverse disease-associated DNA methylation changes and provide protection from joint-destroying autoimmune processes. The use of anti-cancer drugs as arthritis-specific therapeutics is not unprecedented in pharmacology. For example, methotrexate (MTX), one of the most frequently used DMARDs in RA treatment, was originally developed in 1948 to treat childhood leukemia [115], and three decades later, low-dose MTX was introduced as an effective anti-rheumatic drug [116]. We expect that the proposed studies will demonstrate the potential of the methyltransferase inhibitor family in arthritis therapy and facilitate their evaluation in the context of RA.

Histone deacetylases (HDACs) are enzymes that catalyze histone deacetylation and promote chromatin remodelling [117]. Most of the HDACs can interact with DNMTs and be cooperatively involved in gene silencing [118]. The use of HDAC inhibitors (HDACi) as therapeutic drug for autoimmune diseases has been evaluated in preclinical studies [119]. Pan-HDACis were associated with a number of side effects during clinical trials for cancer treatment; therefore, cautious optimism is reasonable regarding of therapeutic potential of this drug family.

Bromodomain (BET) proteins recognize acetylated lysines (characteristic epigenetic activation signal) and are responsible for transmitting this gene activation signal. Accordingly, targeted blocking of BET proteins inhibits the transduction of gene activation (i.e., transcription).

Recently, the inhibition of epigenetic reader proteins, such as the BET proteins, arose as a new therapeutic concept for autoimmune diseases. A BET inhibitor proved to be efficient in a mouse model of autoimmune arthritis by suppressing differentiation and activation of Th17 cells [120]. BET inhibitor worked by suppressing pro-inflammatory cytokines, chemokines, and MMTs in RASF [121]. Currently BET-inhibitor(s) are not among RA therapeutics; however, a number of running clinical trials are evaluating BET inhibitors in oncotherapy [122].

## 7. Conclusions

By employing cutting edge technologies, a number of druggable epigenetic enzymes have been identified and epigenetic mechanisms explored. However, current RA therapies mainly lean on traditional drugs in monotherapy or in combination with rationally designed cytokine neutralizing antibodies. Promising pre-clinical experiments with chromatin-modifying enzyme inhibitors (i.e., DNA and histone modifiers) have been conducted, but still there is no any epigenetic enzyme-targeting drug among RA medications. We firmly believe that after careful evaluation, new epigenetic drugs will be introduced into RA treatment, which can ultimately replace the current DMARDs or act in combination with them to achieve more effective RA treatment.

## Figures and Tables

**Figure 1 ijms-20-05646-f001:**
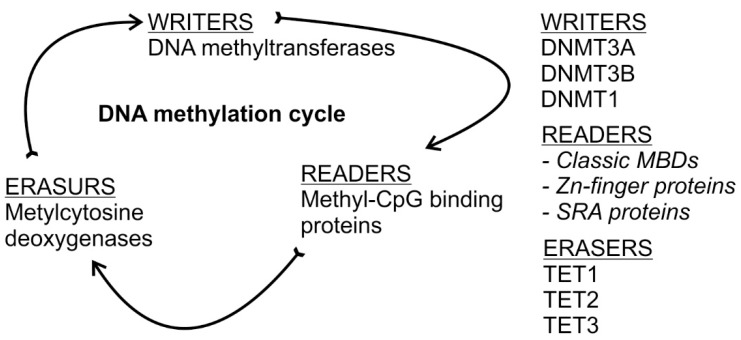
Passing and cycling of epigenetic information through DNA modification. *Left panel* presents process of DNA methylation code writing, decoding and erasing. *Right panel* enlists the involved enzymes.

**Figure 2 ijms-20-05646-f002:**
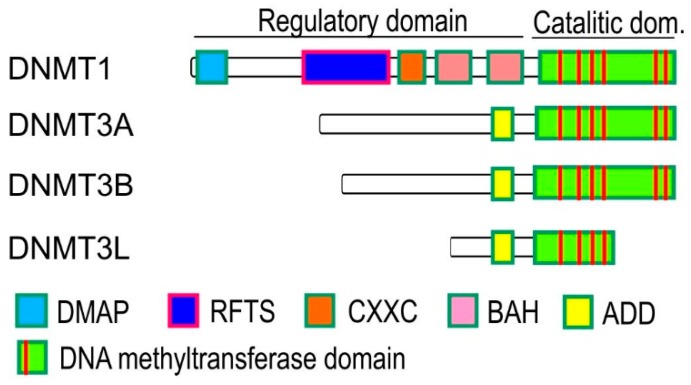
Writers of DNA methylation. Domain structure of DNA methylation code writers. DMAP: DMAP1-binding domain, RFTS: replication foci targeting sequence, CXXC: cysteine-rich Zn^2+^ binding domain, nBAH: Bromo adjacent domain, ADD: ATRX-Dnmt3-Dnmtl domain.

**Figure 3 ijms-20-05646-f003:**
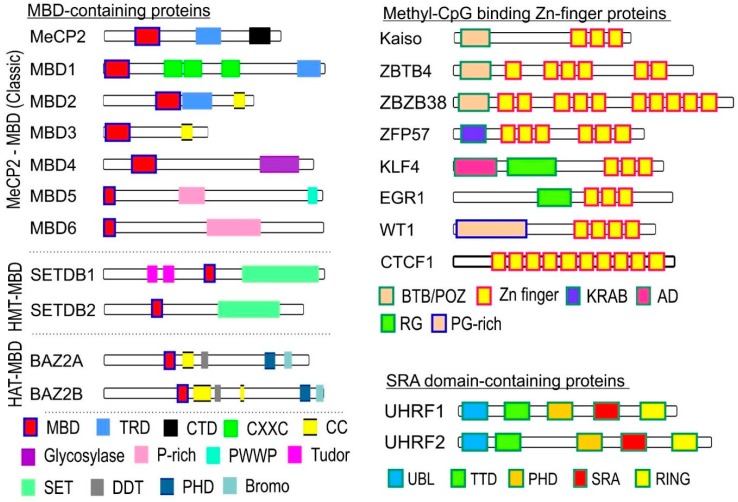
Readers of DNA methylation code. Domain structure of decoding proteins. MBD: Methyl-CpG binding domain, TRD: transcription repression domain, CTD: C-terminal domain, CXXC: cysteine-rich Zn^2+^ binding domain, CC: coiled-coil domain, Glycosylase: DNA glycosylase domain, P-rich: proline-rich domain, PWWP: Pro-Trp-Trp-Prolin motif domain, Tudor: tudor domain, SET: histone methyl transferase domain, DDT: DNA binding homeobox and different transcription factor domain, PHD: plant homebox domain, Bromo: Bromodomain.

**Figure 4 ijms-20-05646-f004:**
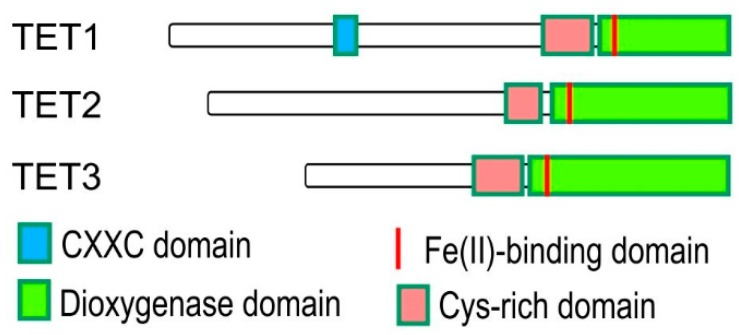
Erasure of DNA methylation. Domain structure of TET proteins. CXXC: cysteine-rich Zn^2+^ binding domain, Fe(II)-binding domain, Dioxygenase domain, Cys-rich domain.

**Figure 5 ijms-20-05646-f005:**
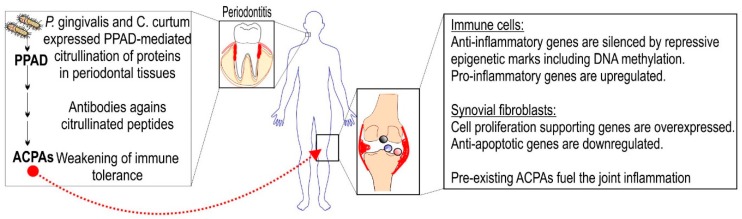
Periodontitis-initiated pathogenesis of RA. Oral microbiota-derived PPADs initiate modification of periodontal tissue proteins that can trigger inflammation in situ and evolve autoimmunity against citrullinated epitopes in synovial joints. Epigenetic signals and mechanisms support RA initiation and progression by generating such an epigenetic milieu that promotes expression of pro-inflammatory genes in immune cells and also fuels high proliferation of synovial fibroblasts.

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
