# Peer review of "DNA Methylation-Governed Gene Expression in Autoimmune Arthritis"

_ijms, 2019, doi:10.3390/ijms20225646_

Round 1

Reviewer 1 Report

Brandt et al. reviewed recent data regarding the epigenetic alterations associated with the pathogenesis of rheumatoid artritis (RA), an autoimmune disease. After summarizing essential and up-to-date background data on DNA methylation, the authors discussed how genome-wide changes in DNA methylation may affect the gene expression pattern of synovial fibroblasts and various immune cells. These cell types play a significant role in the initiation and progression of RA. Because epigenetic alterations are reversible, DNA methylation inhibitors and other drugs targeting various epigenetic regulators may have a therapeutic potential in RA, a topic also outlined by the authors. They also discussed the putative contribution of microbe-induced epigenetic alterations in RA pathogenesis, which is an area of intense research these days. This is a focused review on an important topic. I recommend the publication of the manuscript after a minor revision. I suggest the following changes: line 66, ...enzyme...  change to: ...protein... line 317, ...in OASFs...   change to: in osteoarthritis synovial fibroblasts (OASFs)...  line 317, ...depomethylation...  change to: ...hypomethylation...  line 319, ...rather unique...  change to: ...rather than unique... line 345, ...makes...  change to: ...make...  line 355 and 356, ...Similar DNA methylation pattern changes have been observed in association with RA, wherein global DNA hypomethylation and hypo- or hypermethylation on certain promoters and enhancers [44,81]. - change to:  ...Similar DNA methylation pattern changes have been observed in an experimental model of autoimmune arthritis [44,81].  line 372, ...leaded...  change to: ...led...  line 386, ...finding...  change to: ...findings...  line 389, ...that supervise...  change to: ...regulating...  line 413, ...Porphyromonas gingivalis... - please use italics  line 417, ...members...  change to: member...  line 417, ...translocated in nucleus... change to: ...translocated into the nucleus...  line 424, ..Porphyromonas gingivalis... - please use italics  line 425, ...Cryptobacterium curtum (C. curtum) - please use italics  line 429, ...reduces... change to: ...reduce...  line 429, ...P. gingivalis... - please use italics  line 461, ...It is expected that...  - skip these words.

Author Response

We greatly appreciate the reviewer’s time for evaluating our manuscript and the constitutive comments. We conducted the suggested alterations, which were incorporated into the revised manuscript. We truly believe that the suggested changes and corrections make the manuscript definitely clearer.

Reviewer 2 Report

In this manuscript, Barbara Brandt et al review recent advances on RA-associated methylomes. After a brief introduction, the authors describe epigenetic enzymes by dividing them into 3 groups (writers, readers, and erasers). Then, the authors focus on recent discoveries in the field of RA-associated methylomes, both in RA synoviocytes and in immune cells. They discuss also the growing role of altered microbiota in RA and its impact with epigenetic machinery. Finally, they conclude in the last section with the potential therapeutic applications of epigenetic enzymes targeting drugs.

The paper is clearly written, interesting and timely.

Before publication,  I have a few specific points to address:

From a general point of view, I find that the absence of figures is sorely lacking, in order to give a good overall picture of how epigenetic mechanisms are altered in patients. I think also that the authors should consider developing a bit less the section 2 “writing, interpreting and erasing DNA methylation”. Indeed, I found the trip long enough to finally get to the central part of the review, i.e. epigenetic alteration in RA… once again, I think that a figure illustrating the key actors involved in DNA methylation that are described in sections 2 and 3 would alleviate these two sections.

In section 4, I appreciated that the authors detail the alterations affecting not only immune cells, but also the stromal cells, the synoviocytes. At the beginning of this section, it could be interesting to introduce the concept of early RA which is mainly immune-driven and opposed to the late stage, in which RA may be viewed as a cell-autonomous genetic and epigenetic disease, characterized by altered cell death pathways in synoviocytes after long-term exposure to inflammation. This concept has been detailed in a recent review (Coutant F et al, Current Opinion in Rheumatology, 2019). It would be good to mention it.

Author Response

We greatly appreciate the reviewer’s time for evaluating our manuscript and the constitutive comments. We conducted the suggested alterations including embedding figures, shortening certain sections and adding additional reference. We truly believe that the suggested addition and corrections make the manuscript definitely clearer and more reader-friendly.